# The Relationship between Immunogenicity and Reactogenicity of Seasonal Influenza Vaccine Using Different Delivery Methods

**DOI:** 10.3390/vaccines12070809

**Published:** 2024-07-21

**Authors:** Daniel J. Gromer, Brian D. Plikaytis, Michele P. McCullough, Sonia Tandon Wimalasena, Nadine Rouphael

**Affiliations:** 1The Hope Clinic of the Emory Vaccine Center, Division of Infectious Diseases, Department of Medicine, Emory University, Decatur, GA 30030, USA; m.b.paine@emory.edu (M.P.M.); sonia.tandon@emory.edu (S.T.W.); nroupha@emory.edu (N.R.); 2Laney Graduate School, Emory University, Atlanta, GA 30307, USA; 3BioStat Consulting, LLC, Jasper, GA 30143, USA; bdplikaytis@gmail.com

**Keywords:** influenza vaccine, immunogenicity, reactogenicity

## Abstract

Vaccine immunogenicity and reactogenicity depend on recipient and vaccine characteristics. We hypothesized that healthy adults reporting higher reactogenicity from seasonal inactivated influenza vaccine (IIV) developed higher antibody titers compared with those reporting lower reactogenicity. We performed a secondary analysis of a randomized phase 1 trial of a trivalent IIV delivered by microneedle patch (MNP) or intramuscular (IM) injection. We created composite reactogenicity scores as exposure variables and used hemagglutination inhibition (HAI) titers as outcome variables. We used mixed-model analysis of variance to estimate geometric mean titers (GMTs) and titer fold change and modified Poisson generalized estimating equations to estimate risk ratios of seroprotection and seroconversion. Estimates of H3N2 GMTs were associated with the Systemic and Local scores among the IM group. Within the IM group, those with high reaction scores had lower baseline H3N2 GMTs and twice the titer fold change by day 28. Those with high Local scores had a greater probability of seroconversion. These results suggest that heightened reactogenicity to IM IIV is related to low baseline humoral immunity to an included antigen. Participants with greater reactogenicity developed greater titer fold change after 4 weeks, although the response magnitude was similar or lower compared with low-reactogenicity participants.

## 1. Introduction

A vaccine’s immunogenicity and reactogenicity partially depend on recipient characteristics, including prior antigen exposure. Limited information exists regarding whether vaccine reactogenicity and immunogenicity are related to each other.

Although questions about the association between immunogenicity and reactogenicity had been asked infrequently prior to 2020, the pace of the implementation of SARS-CoV-2 vaccines to address the COVID-19 pandemic reinvigorated inquiry about the immunogenicity–reactogenicity relationship. Several publications featuring large samples, advanced biostatistical methods, and different SARS-CoV-2 vaccines subsequently found positive associations between reactogenicity and immunogenicity, particularly after multiple doses of mRNA vaccines or after participants with prior infection were exposed to vaccine antigen [1,2,3,4,5,6,7,8]. Specific post-vaccination reactions (e.g., fever) and composite scores of reactions were both associated with increased immunity [3,4,5,6,9,10,11,12,13,14,15,16,17]. While the estimation of this relationship for SARS-CoV-2 vaccines represents an exciting step, it is unclear if vaccines against other common respiratory pathogens, such as influenza, carry the same immunogenicity–reactogenicity association.

Over the past several decades, numerous seasonal influenza vaccines have established a clear record of reliable immunogenicity and tolerable reactogenicity. Responses are comparable between split and subunit formulations, all of which contain one hemagglutinin antigen from influenza lineages A/H1N1 and A/H3N2, and one or two antigens from two distinct influenza B lineages [18,19,20]. Healthy adults typically achieve the standard serologic surrogate of seroprotection weeks after vaccination [19], and detectable antibody titers wane over the ensuing 6–12 months [21,22]. Reactogenicity has been extensively characterized, with injection site pain and certain systemic events, particularly headache, myalgia, and malaise, occurring commonly in a mild fashion among young and middle-aged adults receiving intramuscular vaccine [20,23,24]. Intradermal vaccination has resulted in comparable immunogenicity and systemic reactogenicity, with increased local reactogenicity [25].

One major challenge in estimating the immunogenicity–reactogenicity relationship of influenza vaccines comes in the form of confounding by human variation. Even in a clinical trial without variation in the vaccine of interest, heterogeneity in recipient characteristics may impact the estimation of the effect of reactogenicity on immunogenicity. Among multiple SARS-CoV-2 vaccines, younger age, female sex, and prior COVID-19 have all been associated with increases in both immunogenicity and reactogenicity, while the effect of body mass index (BMI) has yet to be shown consistently [2,3,4,6,7,8,26,27]. Age [19,28,29,30] and assigned sex [28,31,32] affect seasonal inactivated influenza vaccine (IIV) responses similarly, although prior antigen exposure via vaccination may be associated with decreased immunogenicity due to imprinting [33,34]. The available data are inconsistent on the topic of the effect of BMI on influenza vaccine responses, perhaps in part due to differences in how BMI is categorized and expressed in various modeling approaches [28,35,36,37,38,39,40]. Although differential vaccine effects are widely recognized and have been for decades, the implications of this heterogeneity in outcomes remains understudied, and the approach to vaccination of large populations often follows a one-size-fits-all model. Seasonal influenza is currently the only pathogen for which we have dedicated vaccines for older (≥65) and younger (<65) adult populations [24,41,42,43], and there is currently no tailored vaccine approach to address differences in sex, BMI, vaccination history, or any other variable. Thus, models investigating the association between reactogenicity and immunogenicity must adjust for multiple confounding variables to achieve accurate estimation.

Heterogeneity in the platform, route, and dose of influenza vaccines represents a second major challenge in estimating the immunogenicity–reactogenicity relationship, as each of these variables can affect both outcomes [19,29,44,45,46,47]. The number and dose of antigen, as well as the presence of an adjuvant, vary among licensed seasonal IIVs, precluding simple comparisons between trial participants who have received vaccines from different manufacturers. One way to circumvent this issue may be to analyze trials of the same vaccine delivered across distinct routes. Although the seasonal IIV is generally offered, as with 93% of FDA-approved vaccines, as an intramuscular (IM) injection via hypodermic needle [45,48], other routes of administration, such as intradermal vaccination via microneedle patch (MNP), are gaining attention. An MNP is an array of sub-millimeter needles attached to a patch backing that delivers antigens to the intradermal space [44,45]. Along with the logistical benefits that MNPs promise, such as low-cost manufacturing, thermostability, easy transportation, lack of waste, increased safety, and possibly the convenience of self-administration [49,50,51,52,53], several preclinical and clinical studies of MNP vaccines have shown equal or enhanced immunogenicity over traditional IM vaccination [49,54,55,56,57]. Because MNPs administer vaccine to the skin, their local reactogenicity is more superficial and visible, resulting in an adverse event profile distinct from their IM counterparts. In general, MNP vaccines cause less pain than IM vaccines, with local reactogenicity instead manifesting as redness, swelling, and itching where the patches are applied [49,56,57,58,59,60].

Therefore, to assess the relationship between reactogenicity and immunogenicity with a seasonal influenza vaccine, we performed a secondary analysis of TIV-MNP 2015, the completed first-in-human phase 1 clinical trial of a seasonal IIV delivered by microneedle patch (MNP), which was compared with intramuscular (IM) injection of the same IIV [49]. Our goal was to estimate the effect of reactogenicity on immunogenicity for each delivery method, adjusting for measured confounding variables.

## 2. Materials and Methods

### 2.1. Parent Study Design

We used primary data obtained in the TIV-MNP 2015 trial [49], a partly blinded, randomized, placebo-controlled, phase 1 study, described in brief below.

### 2.2. Setting and Participants

The TIV-MNP 2015 study included 100 healthy, non-pregnant, immunocompetent adults aged 18–49 years, who were recruited by the Hope Clinic of the Emory Vaccine Center in Atlanta, Georgia in the summer of 2015. Key exclusion criteria included influenza infection or vaccination during the 2014–2015 season, known allergy to egg or other study product components, BMI > 35 kg/m^2^, recent blood donation, vaccination, or experimental product receipt, various acute and chronic medical or psychiatric conditions, and receipt of specified immunosuppressive or immunomodulatory medications. Please see clinicaltrials.gov (accessed on 12 July 2024), NCT02438423, for further details.

### 2.3. Randomization and Procedures

Participants were randomized in a 1:1:1:1 fashion to receive either IIV via microneedle patch (MNP_IIV-HCW_), IIV via intramuscular injection (IM_IIV_), or placebo via microneedle patch (MNP_placebo_), all administered by a healthcare worker, or IIV via microneedle patch self-administered by the study participant (MNP_IIV-self_) under healthcare worker supervision. The IIV was composed of 15 μg of each of the following influenza vaccine strains: A/Christchurch/16/2010, NIB-74 (H1N1), A/Texas/50/2012, NYMC X-223 (H3N2), B/Massachusetts/2/2012, NYMC BX-51 (B). Participants were followed for 180 days. Solicited local and systemic adverse events were assessed daily for 8 days after study product administration, by questionnaire initially and clinic visit if necessary. Adverse events were graded by severity (0 representing no event and 4 representing life-threatening event) based on the Food and Drug Administration toxicity grading schema [61]. Blood samples were drawn for immunogenicity testing on days 0, 28, and 180. We excluded all participants in the MNP_placebo_ group, as they did not have a change in their serologic responses over time due to lack of exposure to an active product and the association of interest was specific to those receiving IIV. Two additional participants did not have any immunogenicity measurements available. We treated these values as missing at random and excluded the immunogenicity data from statistical comparisons. There was no loss to follow-up.

### 2.4. Variables

#### 2.4.1. Reactogenicity

We chose measurements of reactogenicity derived from solicited adverse events as the exposure variables. Reactogenicity was divided into local and systemic solicited adverse events. Local adverse events included swelling (induration), pain, redness (erythema), itching (pruritus), and tenderness. Systemic adverse events included fatigue, joint pain (arthralgia), body ache (myalgia), fever, shivering or shaking body movements, malaise, nausea, sweating, and headache.

#### 2.4.2. Primary Measures of Reactogenicity

The number of unique solicited adverse events recorded by each participant at any point during the 8-day reporting period was summed to make continuous variables. We generated separate sums for local events, systemic events, and all events and termed these Local, Systemic, and Global reaction scores. We examined the distributions of these scores using descriptive statistics and univariable and bivariable plots, visually determined cut points to separate high and low levels of each score, and generated dichotomous categorical variables as our primary reactogenicity measures.

#### 2.4.3. Secondary Measures of Reactogenicity

We grouped participants by the severity of solicited adverse events. Any participant recording an event of grade 2 or greater at any point during the study was included in the high severity group, and all others were included in the low severity group.

We also grouped participants by the duration of solicited adverse events. Any participant with an event beginning on day 0 or day 1 and lasting greater than 2 continuous days (i.e., still present on day 2 or day 3, respectively) was included in the prolonged duration group, and all others were included in the short duration group.

#### 2.4.4. Immunogenicity

We chose measurements of immunogenicity as the outcome variables. Immunogenicity was measured by hemagglutination inhibition (HAI) antibody titer using previously described methods [62].

#### 2.4.5. Primary Measure of Immunogenicity

We chose HAI titers as the primary measure of immunogenicity. HAI titers were represented as continuous numerical data. They were log-transformed for regression analyses and then back-transformed to generate geometric mean titers (GMTs) and the ratios between GMTs (geometric mean titer ratios, or GMRs).

#### 2.4.6. Secondary Measures of Immunogenicity

We modeled HAI titer fold change as a measure of immunogenicity. We calculated titer fold change by subtracting baseline log-transformed HAI from post-vaccine (day 28 or day 180) log-transformed HAI for each participant.

We also chose HAI seroprotection and HAI seroconversion as dichotomous categorical measures of immunogenicity. Seroprotection is defined as an HAI titer ≥1:40. Seroconversion is defined as a post-vaccination measurement with a minimum 4-fold increase in HAI titer (if the baseline titer is ≥1:10) or an HAI titer ≥1:40 (if the baseline titer is <1:10).

#### 2.4.7. Covariates

Other measured covariates, recorded at enrollment, included BMI, sex, race, ethnicity, and IIV receipt in the prior two influenza seasons. All covariates were recorded by participant questionnaire except for BMI, which was measured. All were represented as categorical variables. BMI was categorized as ≤25, 25 to <30, and ≥30 kg/m^2^.

### 2.5. Sensitivity Analysis

To assess the robustness of findings generated with our primary measures of reactogenicity, we repeated our analyses using multiple different cut points for dichotomizing reaction scores. Additionally, we divided the participants into those with the 15 highest and 15 lowest values of each reaction score and performed comparisons between these groups of participants. For example, we compared those with the 15 highest Global scores to those with the 15 lowest Global scores.

### 2.6. Statistical Analysis

We summarized the exposure, outcome, and covariates using descriptive statistics and plots. We used histograms to display the frequency of solicited adverse events and constructed a correlation matrix using Pearson’s correlation coefficients and hierarchical clustering to assess for symptom clustering. To further characterize potential confounders and effect modifiers, we performed bivariable comparisons between selected variables and both the exposure and outcome measures. Covariates with statistically significant associations with both the exposure and outcome at the 95% confidence level were marked as potential confounders for adjustment in the multivariable analyses. Additionally, we assessed interactions in multivariable models.

For the primary multivariable analyses, we used mixed-model analysis of variance (ANOVA) to estimate GMTs and GMRs, with the exposure and time as independent variables and employing a variance component covariance structure. We included vaccine delivery method (IM or MNP) in interaction terms with these factors, including a three-way interaction term, to generate separate estimates for participants receiving IM or MNP IIV on each study day. We used analogous methods for models estimating titer fold change, which were expressed as ratios between groups (fold change ratios, or FCRs). For analyses with seroprotection and seroconversion outcomes, we used modified Poisson regression [63,64] and generalized estimating equations (GEEs) [65,66] to generate risk ratios (RRs).

All statistical testing was performed in SAS version 9.4 (Cary, NC, USA). Longitudinal models were created to account for repeated measures. We used the MIXED procedure to generate estimates with mixed-model ANOVAs and the GENMOD procedure to generate estimates with modified Poisson regression and GEEs. Data visualization was performed with R v4.2.1 (R Foundation for Statistical Computing, Vienna, Austria).

## 3. Results

The phase 1 trial randomized 100 participants (Figure 1). After the 25 participants in the MNP_placebo_ group were excluded, we excluded the missing immunogenicity data of two additional participants. We combined the MNP_IIV-HCW_ and MNP_IIV-self_ groups into a single MNP group to compare with the IM group. The two groups were balanced in baseline characteristics (Appendix A). The median baseline H1N1 HAI titer was 80, and the median H3N2 and B HAI titers were 40.

Appendix A displays the fraction of participants who received either IM or MNP IIV and had each type of reaction. The IM group reported more systemic events of every type except fever, which was only reported by a single participant, who was in the MNP group. The MNP group reported more local pruritus, redness, and swelling events, and the IM group reported more local pain events. Redness and swelling peaked within 24 h of vaccination and resolved promptly, although earlier in the IM group (Appendix A).

Figure 2 displays the correlation matrix between all types of solicited adverse events. Most Pearson correlation coefficients were close to 0, and there was no discernible hierarchical clustering pattern between events.

Based on the distributions of the Global, Systemic, and Local reaction scores generated for all participants, we visually dichotomized the scores into high and low levels (Appendix A). Those with a Global reaction score ≥4, a Systemic reaction score ≥3, or a Local reaction score ≥3 were assigned the high level for the corresponding score.

To screen for significant associations between reactogenicity and immunogenicity, we generated unadjusted and adjusted longitudinal GMR estimates and hypothesis tests for each vaccine antigen comparing participants in the IM or MNP groups with high and low Global, Systemic, and Local reaction scores (Table 1). We found no significant associations using the Global score but noted a consistent association among the IM group between HAI titers against the H3N2 antigen and both the Systemic and Local scores. We also noted associations between the Local score and the H1N1 and H3N2 HAI titers in the MNP group. These associations were not robust to changes in the dichotomization cut points of the reaction scores (Appendix A–C).

When we used interaction terms to generate time-specific estimates (Appendix A–C), we found that the previously noted associations among the MNP group were not driven by any significant difference on any specific day. In contrast, the IM group associations were driven by baseline differences in H3N2 HAI titer (reproduced from Appendix A in Table 2). Among participants in the IM group, adjusting for BMI, sex, race, ethnicity, and prior IIV, those with a high Systemic reaction score had 0.4 (95% CI 0.2, 0.8) times the baseline H3N2 GMT compared with those who had a low Systemic score. Similarly, those with a high Local score had 0.3 (95% CI 0.1, 0.8) times the baseline H3N2 GMT compared with those who had a low Local score. Those in the IM group with a high Systemic score developed similar H3N2 GMTs on day 28 compared with those who had a low Systemic score. In contrast, those with a high Local score appeared to develop lower H3N2 GMTs on day 28 compared with those who had a low Local score, although this did not reach statistical significance. Finally, those with high Systemic and Local scores appeared to demonstrate lower H3N2 GMTs on day 180 compared with their low-score counterparts, although these findings also did not reach statistical significance (Table 2, Appendix A).

We generated analogous unadjusted and adjusted estimates for the secondary outcome of HAI titer fold change (Appendix A–C). Among the IM group, prior to adjustment for confounding variables, those with any type of high reaction score had more than 2 times the day 28 H3N2 titer fold change compared with their low-score counterparts. This finding remained similar after adjustment and was statistically significant when using the Global reaction score (reproduced from Appendix A in Table 3). Although it was not indicated by the longitudinal GMR estimates, we found a similar pattern among the MNP group using the Global reaction score and the B antigen, where a high score was associated with lower baseline HAI titer (Appendix A) and greater day 28 titer fold change (Appendix A).

As a sensitivity analysis, we selected participants with the 15 highest and 15 lowest values of each reaction score and performed unadjusted comparisons between these smaller groups of participants with extreme values. Among the IM group, those with the highest Local scores had lower H3N2 GMTs compared to those with the lowest Local scores (Appendix A), and those with the highest Systemic scores had a higher H3N2 titer fold change on day 28 compared to those with the lowest Systemic scores (Appendix A). Additionally, those in the IM group with the highest Local reaction scores had higher H1N1 GMTs and a lower H1N1 titer fold change on day 28 compared to those with the lowest Local scores, but this association was not found in the primary analyses.

We then grouped participants by either reaction severity or reaction duration, agnostic of reaction type, and estimated the relationship between these secondary reactogenicity measures and either HAI GMT or titer fold change. We found no association between moderate or greater reaction severity and either longitudinal GMT (Appendix A) or titer fold change (Appendix A). We also found no association between a “prolonged” reaction duration greater than 48 continuous hours and either longitudinal GMT (Appendix A) or titer fold change (Appendix A).

Using modified Poisson GEEs, we then generated unadjusted and adjusted longitudinal estimates for the secondary outcome of HAI seroprotection (Appendix A). There were no significant associations between any reaction score and seroprotection, and models generating time-specific estimates frequently failed to converge due to the high proportion of participants with seroprotection, including at baseline (Appendix A). There were also no significant associations between any reaction score and seroprotection after changing the dichotomization cut points of the scores (Appendix A).

Similarly, we generated unadjusted and adjusted longitudinal estimates for the secondary outcome of HAI seroconversion (Appendix A), noting only an association between the Systemic reaction score and H3N2 seroconversion in the MNP group after adjustment for confounding variables. When we generated adjusted time-specific estimates (Appendix A–C), we found that those with a high Local score had 1.4 (95% CI 1.0, 1.9) times the risk of day 28 H3N2 seroconversion compared to those with a low Local score among the IM group (Appendix A). Those with a high Systemic score had 2.9 (95% CI 1.4, 6.0) times the risk of day 180 H3N2 seroconversion compared to those with a low Systemic score. These findings were not robust to changes in the dichotomization cut points of the reaction scores (Appendix A).

## 4. Discussion

In this secondary analysis of a phase 1 clinical trial comparing the same seasonal IIV when delivered by IM injection or MNP, we found evidence of a relationship between reactogenicity and immunogenicity in the IM group. Our results suggest that heightened reactogenicity to IM IIV is related to low baseline HAI antibody titers to included antigens, in this case, H3N2. Participants with greater reactogenicity developed greater H3N2 HAI titer fold change after 4 weeks, although the overall magnitude of HAI response was similar or lower compared with low-reactogenicity participants. For those with relatively greater local reactogenicity, who appear to have had the lowest baseline H3N2 HAI titers, the greater titer fold change was associated with a greater probability of seroconversion.

These findings appear to conflict with the SARS-CoV-2 vaccine literature, in which participants without prior infection reported greater reactogenicity and developed most of their immunity from the second (boost) dose of a prime-boost vaccine series. Our results instead show that those with greater reactogenicity had lower baseline markers of humoral immunity. The antibody response of these participants in the short term was higher in the relative sense and similar or lower in the absolute sense than their low-reactogenicity counterparts. The reasons behind these differences between our expectations and our findings are not clear. However, investigating the phenomenon behind the apparently complex relationship between reactogenicity and immunogenicity is of critical importance. Understanding this relationship can potentially define a clinical indicator of vaccine efficacy in the early stages of a pandemic emergency and help clinicians to counsel patients about the meaning behind post-vaccine adverse events. It is a necessary step toward designing vaccines and adjuvants that maximize both tolerability and immunity and may even lead to connections between typical vaccine responses and rare adverse events of special interest, such as myocarditis and Guillain-Barré syndrome.

Regarding why those with lower baseline HAI titers might have greater reactogenicity, one possibility is that those with greater preexisting immunity do not undergo the same inflammatory cascade as those with less preexisting immunity, perhaps due to antibody-mediated precipitation and destruction of injected antigen without optimal presentation by APCs in the draining lymph node. This could also explain why the difference was only noted in our study in the IM group, although the MNP group was larger and had more statistical power—the muscle has a sparse distribution of APCs and lymphatic vessels compared to the dermis [54,67].

A complicating factor in this analysis that might explain why participants had differing levels of baseline H3N2 HAI titers is that we are unable to accurately discern each participant’s personal history of antigen exposure. We do not have granular data about trial participants’ seasonal IIV and influenza infection histories, including what antigens and strains they have been exposed to, how many times, in what order, and how recently. All these pieces of information may affect a person’s baseline HAI titer to specific antigens, such as the A/Texas/50/2012, NYMC X-223 used as the H3N2 component in the TIV-MNP 2015 study. We adjusted for receipt of influenza vaccine within the prior 2 years, as recent vaccination may affect responses to a new influenza vaccine via imprinting, but we were unable to adjust for most influenza antigen exposure in the participants’ lifetimes. The mismatch of H3N2 vaccine antigen and influenza strain from the preceding influenza season [68,69] may also have been important here, as having had undiagnosed influenza A/H3N2 during the prior winter would not have primed participants perfectly for the H3N2 antigen they were exposed to in the trial.

This may also help to explain why these models only generated consistent and significant findings with the H3N2 HAI titer outcome. The H1N1 antigen used in influenza vaccines over the previous 5 years, including in a standalone pandemic influenza vaccine, had not changed [68,70]. Individuals exposed to pandemic A/H1N1 influenza in 2009, vaccinated against this strain specifically during the pandemic emergency, or vaccinated against seasonal influenza between 2010 and 2014 were effectively primed with the same H1N1 antigen used in the TIV-MNP 2015 trial. This is supported by our findings that median baseline H1N1 HAI titers were higher than H3N2 or B HAI titers and that over 75% of participants had baseline seroprotection against A/H1N1 influenza.

Why we did not find a consistent association between B HAI titer and reactogenicity is harder to rationalize. The median baseline HAI titer was the same against H3N2 and B antigens, and the percentage of participants with baseline seroprotection was lower for the B antigen than for the H3N2 antigen. However, the IM group had higher baseline seroprotection against influenza B than against influenza A/H3N2, and the MNP group had lower baseline seroprotection against influenza B. Consistent with this, among the MNP group, a high Global reaction score was associated with lower baseline B HAI titer and a higher day 28 B HAI titer fold change. Despite this signal, we must exercise caution when interpreting influenza B HAI results from a study of a trivalent vaccine for the same reasons discussed above. Many participants most likely had unique personal histories of influenza B antigen and virus exposure, leading to heterogeneity in their baseline HAI titers and their anamnestic responses to vaccination with an influenza B antigen from one of two relatively stable co-circulating lineages [71,72].

Taken together, it is challenging to compare IIV data to SARS-CoV-2 vaccine data. The COVID-19 pandemic resulted in a novel pathogen exposure and novel vaccines, leading to repetitive antigen exposures over a short period of time. SARS-CoV-2 vaccine study participants did not have long histories of intermittent exposure to several sequences of antigen at varying ages and periods of immune system development, as is the case with influenza. It is possible that future data on the immunogenicity-reactogenicity relationship of SARS-CoV-2 vaccines, derived from populations with hybrid immunity and diverse antigen exposures over several years and at different stages of life, may reflect a phenomenon similar to the one we observed in this study. One early investigation into this topic showed a reduction in reactogenicity between the primary SARS-CoV-2 vaccine series and homologous booster doses, and a meta-analysis comparing homologous and heterologous boosting demonstrated a higher risk of nonserious adverse events with heterologous boosting. Both results suggest a negative association between preexisting homologous immunity and vaccine-related adverse events [73,74]. Additionally, a newly published H7N9 influenza vaccine study showed a reduction in grade 3 injection site reactions between the prime and boost doses [75]. Future early-phase clinical trials of both influenza and SARS-CoV-2 vaccines should attempt to gather, if possible, more granular antigen exposure histories.

Although HAI titer remains a standard correlate of protection for FDA approval of vaccines against influenza, the relationship between HAI and reactogenicity does not account for other measures of humoral immunity, including neuraminidase inhibition (NAI) antibody titers, or measures of cellular immunity, such as cytokine levels and specific T cell populations. This is a key limitation of our study (and many others in this space). Another limitation is our a priori approach to defining reactogenicity variables as composite scores. These unweighted scores assume that each type of adverse event has the same impact on immunogenicity (e.g., fatigue and myalgia carry the same amount of meaning). However, we also recognize that adverse event grades are highly subjective. Summing grades of events instead of numbers of events could have introduced additional subjectivity to our models. Moreover, using sums of grades injects another assumption: that individual event grades (e.g., headache grades 1, 2, and 3) are equal distances apart. Ultimately, we chose our primary measure of reactogenicity because it is generalizable to future research, interpretable, and can be implemented clinically. Creating a bespoke, weighted composite reactogenicity score in future studies, for example, by using LASSO regression techniques or unsupervised learning methods to find a smaller number of more important solicited adverse event types, could be instructive. However, given the small sample size in this study, such an approach would likely forfeit generalizability to larger databases, if it were feasible at all. Our approach is supported by multiple publications in the SARS-CoV-2 literature, in which numbers of solicited adverse events were added together to define groups of participants or variables for modeling. Although there is no metric to confirm the construct validity of our score definition or our dichotomization cut points, the fact that the findings were consistent and robust to at least some sensitivity analyses lends credence to our chosen approach.

The small sample size, large number of variables, and multitude of model outputs collectively represent an additional limitation of the study. Incorporating longitudinal data into mixed-model ANOVAs and GEEs mitigated the statistical power limitations and prevented additional statistical comparisons to some extent, but it remained challenging to prioritize and interpret model outputs. In some cases, particularly those involving the relative variables of titer fold change and seroconversion, using longitudinal models may have reduced the clarity of the findings. As an example, day 180 seroconversion carries less meaning than day 28 seroconversion, and its inclusion in GEEs obscured the association between local reactogenicity and seroconversion in the longitudinal estimates. At the same time, this study accomplished the function of a screening experiment successfully. We used advanced biostatistical methods on a small sample to identify signals and generate hypotheses for future investigation. One more limitation is the low generalizability of our sample, which only included young and middle-aged adults without a wide range of self-reported racial and ethnic diversity. Although race and ethnicity should not have biologic significance regarding vaccine responses, we still adjusted for these variables due to associations we found in our preliminary bivariable comparisons and because they may act as proxies for unmeasured confounding variables that do affect vaccine biology. Future analyses of the influenza immunogenicity–reactogenicity relationship will need to include a more generalizable sample of healthy participants. Overall, our novel findings will need to be validated using a less exploratory analytic structure and a larger sample.

In conclusion, greater reactogenicity of intramuscular seasonal inactivated influenza vaccine was associated with lower baseline H3N2 immunity and greater antibody titer fold change after 28 days, although the overall magnitude of the response was similar or lower compared with low-reactogenicity participants. Future studies, both secondary analyses of clinical trial outcome data and those employing systems biology approaches, can help us better understand the associations between vaccine responses and their molecular underpinnings. The immunogenicity–reactogenicity relationship has wide-ranging implications for patient care, public health, and vaccine development.

## Figures and Tables

**Figure 1 vaccines-12-00809-f001:**
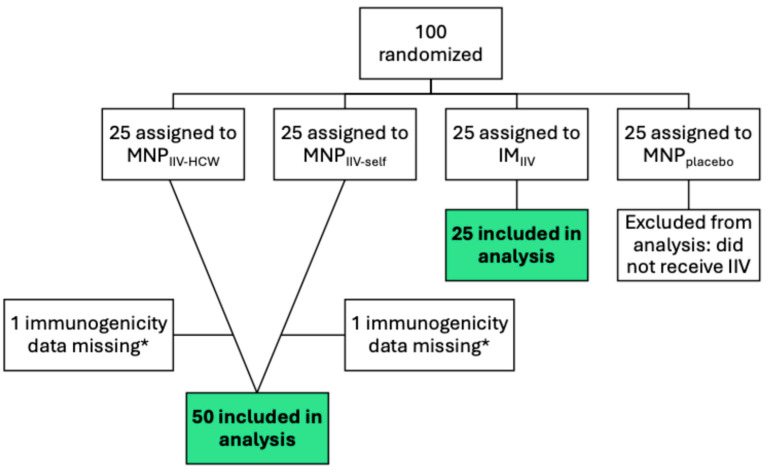
Participant Flow Diagram. Adapted from TIV-MNP 2015 [49] for secondary analysis. * Reactogenicity and covariate data of participants with missing immunogenicity data were included. Green boxes with bold type represent the participants included in the sample for analysis.

**Figure 2 vaccines-12-00809-f002:**
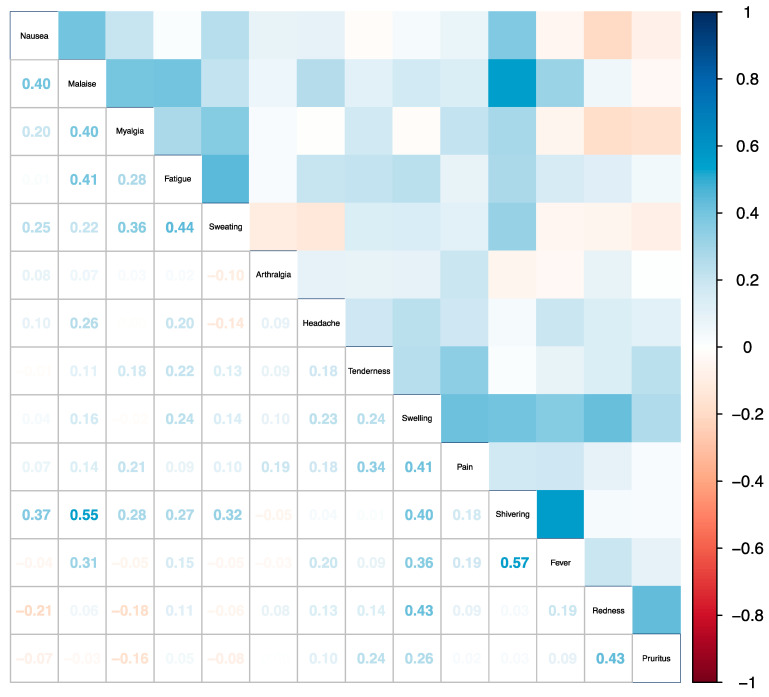
Correlation Between Adverse Events. Correlation matrix of all solicited adverse event types, ordered by hierarchical clustering. Pearson’s correlation coefficients between any two solicited adverse event types are presented on the bottom left and mirrored by a color gradient ranging between 1 (blue) and −1 (red) on the top right.

**Table 1 vaccines-12-00809-t001:** Longitudinal Associations Between Reaction Score and HAI GMT.

Reaction Type	Antigen	Study Group	Unadjusted GMR * (95% CI)	*p*-Value	Adjusted GMR *^ (95% CI)	*p*-Value
Global	H1N1	IM	1.07 (0.53, 2.16)	0.85	0.98 (0.58, 1.66)	0.95
		MNP	0.87 (0.50, 1.52)	0.62	0.85 (0.58, 1.24)	0.38
	H3N2	IM	0.83 (0.47, 1.49)	0.54	0.74 (0.45, 1.22)	0.23
		MNP	0.81 (0.50, 1.29)	0.37	0.78 (0.54, 1.12)	0.17
	B	IM	1.21 (0.80, 1.83)	0.36	0.95 (0.63, 1.42)	0.79
		MNP	0.80 (0.53, 1.22)	0.30	0.81 (0.58, 1.14)	0.22
Systemic	H1N1	IM	0.73 (0.35, 1.52)	0.40	0.65 (0.38, 1.12)	0.12
		MNP	0.91 (0.36, 2.27)	0.83	0.85 (0.45, 1.60)	0.61
	H3N2	IM	0.64 (0.35, 1.18)	0.15	**0.54** **(0.33, 0.89)**	**0.02**
		MNP	1.43 (0.66, 3.10)	0.36	1.42 (0.77, 2.60)	0.26
	B	IM	1.07 (0.69, 1.65)	0.77	0.81 (0.53, 1.24)	0.33
		MNP	0.84 (0.42, 1.66)	0.61	0.78 (0.45, 1.36)	0.38
Local	H1N1	IM	0.59 (0.22, 1.54)	0.27	0.6 (0.30, 1.21)	0.15
		MNP	1.53 (0.90, 2.61)	0.12	**1.54** **(1.05, 2.25)**	**0.03**
	H3N2	IM	0.46 (0.21, 1.01)	0.05	**0.37** **(0.19, 0.72)**	**<0.01**
		MNP	0.72 (0.46, 1.13)	0.15	**0.63** **(0.44, 0.90)**	**0.01**
	B	IM	1.06 (0.60, 1.87)	0.85	0.89 (0.51, 1.55)	0.68
		MNP	0.97 (0.65, 1.45)	0.89	1.00 (0.71, 1.41)	0.99

HAI: hemagglutination inhibition; GMT: geometric mean titer; GMR: geometric mean titer ratio; CI: confidence interval; IM: intramuscular; MNP: microneedle patch. * Estimated geometric mean titer ratio for those in the higher score category compared with those in the lower score category. ^ Adjusted for body mass index category, sex, prior inactivated influenza vaccination, race, ethnicity, and study day. Bolded type indicates *p*-Value < 0.05.

**Table 2 vaccines-12-00809-t002:** IM Study Group Reaction Score and H3N2 HAI GMT by Study Day.

Reaction Type	Day 0 Adjusted GMR *^ (95% CI)	*p*-Value	Day 28 Adjusted GMR *^ (95% CI)	*p*-Value	Day 180 Adjusted GMR *^ (95% CI)	*p*-Value
Global	0.5 (0.2, 1.0)	0.06	1.0 (0.4, 2.2)	0.95	0.9 (0.4, 2.1)	0.85
Systemic	**0.4** **(0.2, 0.8)**	**0.01**	0.9 (0.4, 1.9)	0.7	0.5 (0.2, 1.2)	0.12
Local	**0.3** **(0.1, 0.8)**	**0.01**	0.6 (0.2, 1.6)	0.28	0.4 (0.1, 1.1)	0.07

IM: intramuscular; HAI: hemagglutination inhibition; GMT: geometric mean titer; GMR: geometric mean titer ratio; CI: confidence interval. * Estimated geometric mean titer ratio for those in the higher score category compared with those in the lower score category. ^ Adjusted for body mass index category, sex, prior inactivated influenza vaccination, race, ethnicity, and study day. Bolded type indicates *p*-Value < 0.05.

**Table 3 vaccines-12-00809-t003:** IM Study Group Reaction Score and H3N2 HAI Titer Fold Change by Study Day.

Reaction Type	Day 28 Adjusted FCR *^ (95% CI)	*p*-Value
Global	**2.0** **(1.0, 3.7)**	**0.04**
Systemic	1.9 (1.0, 3.8)	0.07
Local	2.0 (0.8, 4.9)	0.15

IM: intramuscular; HAI: hemagglutination inhibition; FCR: fold change ratio; CI: confidence interval. * Estimated fold change ratio for those in the higher score category compared with those in the lower score category. ^ Adjusted for body mass index category, sex, prior inactivated influenza vaccination, race, ethnicity, and study day. Bolded type indicates *p* < 0.05.

## Data Availability

The raw data supporting the conclusions of this article will be made available by the authors on request.

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
