# Peer review of "The Relationship between Immunogenicity and Reactogenicity of Seasonal Influenza Vaccine Using Different Delivery Methods"

_vaccines, 2024, doi:10.3390/vaccines12070809_

Round 1

Reviewer 1 Report

Comments and Suggestions for Authors

This study inspects the relationship between the immunogenicity and the reactogenicity of seasonal influenza vaccines administered by 2 different routes of immunization (IM or MNP). By analyzing the participants' adverse reaction scores and their geometric mean titers of haemagglutination inhibition tests before and after vaccination, the study examines how response amount and type (mild, moderate, or severe) and reaction duration can affect the immune response. This study is an approach to enhance vaccine research and development, optimization, and clinical application with scientific evidence. The outcomes indicate that enhanced reactogenicity, lower baseline immunity, and more enhanced antibody response may indicate a correlation between immunogenicity and reactogenicity. The main points highlighted in the study as below.

1、 The review of related literature didn't articulate the immunogenicity and reactogenicity after seasonal influenza vaccine adequately.

2、 Although the heightened reactogenicity to those with lower pre-existing antibody levels was shown in the results section, it lacks sufficient literature citations or experimental validation to support this hypothesis.

3、 Although the small sample size and the difficulty of investigating past medical history and vaccination history were mentioned in the discussion section, it is worth considering proposing corresponding measures to address these issues.

Reviewer 2 Report

Comments and Suggestions for Authors

The peer-reviewed work examined the relationship between reactogenicity and immunogenicity of trivalent influenza vaccine when administered intramuscularly (IM) and via microneedle patch (MNP). Data obtained from the blinded, randomized, placebo-controlled trial included 100 healthy, immunocompetent adults were subject to secondary analysis to identify correlations between reactogenicity and immunogenicity. Reactogenicity was assessed using composite reactogenicity scores, and immunogenicity was assessed using hemagglutination inhibition (HAI) titers. It has been shown that with intramuscular injection such reactogenicity indicators as pain, myalgia, sweating and malaise predominate, while after MNP local tenderness and pruritus predominates.

 It has been shown that those with a high systemic and local reactions had lower baseline H3N2 GMT compared those with low reactions Systemic score. The “discussion” section analyzes why this phenomenon is characteristic only of the H3N2 immune response, and is not observed for immune responses to H1N1 and B antigens. The article is well written and can be accepted in present form.

Reviewer 3 Report

Comments and Suggestions for Authors

Even though influenza vaccinations have been available for more than 50 years, little research has been given to how the immunization route affects the immune response or the frequency of side effects. The size of the needles that are either suggested or included with the vaccine varies, as do the recommendations about the vaccination route. Few published trials provide comprehensive information about the vaccination site, administration method, and needle length.

The manuscript presented by  Gromer et al. for review,  examines the influence of the route of vaccination, the immune response, and the rate of adverse reactions, making it relevant and interesting for a reader.

However, I have several major concerns about the study design.

First of all, they used a few people in the design of the experiment.

I don't see that they followed up with the individuals with egg protein allergies who should be excluded from the analysis.

The results are presented with overly complicated graphs that are not well explained, making it difficult to understand.

Why do you exclude the placebo group in a serological assay prior to vaccination?

Why did you decide to introduce three different forms of reaction - Local, Systemic, and Global reaction scores, especially since your studied cohort is small?

The research was conducted in 2015, what were the reasons for not reporting it until now?

My opinion is that you need to simplify the way the results are presented so that they are easy to understand and follow. All tables and figures need better explanation and full legend of abbreviations used.

Table 2 Geometric mean titer of haemagglutination inhibition antibodies pre/post vaccination, fold increase pre/post vaccination, percentage with post-vaccination titre >40, percentage 4 with >4-fold increase post-vaccination by route of administration .....

Minor comments

L23 – IIV? It needs to be explained

What is your reason for including the race and ethnicity characteristics in Table 1?

Figure 1 – what is a fraction of participants, how do you calculate the fraction?

Round 2

Reviewer 3 Report

Comments and Suggestions for Authors

My opinion is that the results presented in the article largely repeat the results of the article: The safety, immunogenicity, and acceptability of inactivated influenza vaccine delivered by microneedle patch (TIV-MNP 2015): a randomized, partially blinded, placebo-controlled, phase 1 trial. Lancet.

Table 1 and Figure 2 repeat Table 1 and Figure 3 already presented in the Lancet journal. I think you should consider how much of the previously reported results you should present in this article.

In Table 1, I believe that race and ethnicity should not necessarily be included as characteristics.

L22, 23 – IIV? It needs to be explained, please describe the flu strains
